# Community Perspectives and Environmental Justice Issues in an Unincorporated Black Township

**DOI:** 10.3390/ijerph19127490

**Published:** 2022-06-18

**Authors:** Teriana Moore, Pamela Payne-Foster, JoAnn S. Oliver, Ellen Griffith Spears, Christopher H. Spencer, Jacqueline Maye, Rebecca S. Allen

**Affiliations:** 1Departments of Political Science and Gender and Race Studies and Sociology, The University of Alabama, Tuscaloosa, AL 35487, USA; tmmoore6@crimson.ua.edu; 2Institute for Rural Health Research, Community Medicine and Population Health, The University of Alabama, Tuscaloosa, AL 35487, USA; pfoster@ua.edu; 3Alabama Research Institute on Aging, The University of Alabama, Tuscaloosa, AL 35487, USA; joliver@ua.edu; 4Capstone College of Nursing, The University of Alabama, Tuscaloosa, AL 35487, USA; 5American Studies and New College, The University of Alabama, Tuscaloosa, AL 35487, USA; egspears@ua.edu; 6Black Belt Community Foundation, Selma, AL 36702, USA; cspencer@blackbeltfound.org; 7Holt Project Advisory, Holt, AL 35404, USA; jacqueline.maye@ua.edu; 8Department of Psychology, The University of Alabama, P.O. Box 870348, Tuscaloosa, AL 35487, USA

**Keywords:** industrial pollution, low-income, Black people, environmental justice

## Abstract

Through each era, the southeastern United States was and continues to be an epicenter for industrial companies to establish factories and plants. Though this development attracts economic gain for the companies and surrounding areas, low-income and predominantly Black communities bear the brunt of the environmental consequences while frequently remaining stagnant economically. This qualitative, community-based participatory research study grew out of a larger study designed to recruit lay community advisors from communities labeled as hard to reach in research. We focus on Holt, Alabama, an unincorporated community in the southeastern United States region. The primary goal of this research inquiry is to thematically analyze community interviews stemming from a topic of research, practice, and policy interest to community members: the effects of industrial pollution on Holt citizens’ daily lives. Content analysis of focus-group transcripts revealed four emergent themes, including: (1) how the pollution affects their water, soil, and air quality; (2) illness related to pollution; (3) community engagement and empowerment; and (4) suggestions regarding what government officials could do to address this area of need. Building upon the prior research regarding environmental justice, human flourishing, and the definition of nurturing environments, suggestions are made regarding the creation, implementation, and maintenance of project advisory councils focused on issues of environmental justice. Community advocacy and empowerment as well as community and scientific partnerships are imperative to alleviate problems associated with environmental justice.

## 1. Introduction

Using nationally representative 1986 baseline data from the Americans’ Changing Lives survey, Mohai and colleagues [1] found that Black people and individuals with lower levels of education were more likely to live within a mile of a polluting facility. Geographically, this finding was most pronounced in metropolitan areas of the Midwest and West, as well as suburban areas of the South. Although there is no official United States (US) Census Bureau definition, the US Geological Survey considers the following states as “southeastern”: Alabama, Florida, Georgia, Arkansas, Kentucky, Louisiana, Mississippi, North Carolina, South Carolina, and Tennessee (Puerto Rico and the US Virgin Islands also are included). Given the data reported by Mohai and colleagues [1] is now over 35 years old, the current qualitative inquiry investigated perceived environmental needs from concerns raised directly by community residents in an unincorporated suburban township in the southeastern United States geographic region.

The Holt, Alabama, community was founded in 1901 by industrialist Frank Holt, who was intent on expanding the iron and coal industry to this area. As the coke ovens flourished, Holt had houses built in proximity to the facility to accommodate their employees and their families [2]. Now, 120 years after being established, Holt is a predominantly Black community with 33.1 percent of its residents living in poverty in comparison to the 16.9 percent in the state of Alabama [3]. Living in an unincorporated city, residents often lack a political voice, have depreciated resources, and limited access to government officials to voice community needs. Overcoming the previous lack of knowledge of the presence of pollution or its harmful effects, current residents are realizing the extent of the continuous exposure to pollutants and how that exposure has affected their health and daily quality of life. This awareness reflects current trends in environmental justice research [4], as well as known environmental racial disparities in the southeastern US region [5,6].

In the late 1980s, sociologist Robert Bullard, among other activists, exposed the disproportionate number of pollutant-producing factories and landfills placed in low-income, predominantly Black/Latinx communities. These neighborhoods are called fence-line communities. A fence-line community is a neighborhood that is immediately adjacent to a company and is directly affected by the noise, odors, chemical emissions, traffic, parking, and operations of the company. A 2017 NAACP report noted that African Americans in the US are exposed to 38 percent more polluted air than White Americans while accounting for 75 of fence-line communities [7,8]. Notably, these more recent data coincide with those reported by Mohai and colleagues in the 1986 baseline data from Americans’ Changing Lives [1]. Unfortunately, because this industrial pollution is linked to poor health outcomes, such as asthma, dysfunctional lungs, cardiovascular and respiratory illnesses, and low birth weight for babies, environmental justice remains an important issue for communities of color [9]. Although numerous research studies document the correlation between race, class, and the effects of pollution exposure, decision makers have been unresponsive, making few changes in the siting or regulation of noxious industries [5,6,10].

The current study highlights these challenges, as well as explores potential solutions to address environmental justice from Holt community member’s perspectives. Toxic pollution may have been a particular concern for Holt residents because a nearby coke plant, Empire Coke, a branch of Birmingham-based McWane, Inc., prepared coal for the manufacture of steel. This plant has faced more than one lawsuit over their operations in Holt. McWane officials claimed that the plant operated safely [11]. McWane, Inc. had agreed in 2010 with the US Environmental Protection Agency (EPA), the Justice Department, and the states of Alabama and Iowa to pay USD 4 million and more than twice that amount in cleanup costs to settle 400 violations of federal and state environmental laws at its manufacturing facilities, including the Empire Coke site [12]. The Holt community also lies downstream of several coal mining operations that reportedly discharge into the Black Warrior River, a condition that continued to exist in 2021 [13]. The EPA’s Toxic Release Inventory identified other emitters in the Holt area, a steel plant and a roofing company [14].

This study represents one component of the Patient-Centered Outcomes Research Institute (PCORI)-sponsored Project SOAR (Sharing Opinions and Advice about Research in the Deep South, PCORI Contract #1097). Project SOAR was a community engagement project designed to empower community residents from communities labeled hard to reach in research [15] to work with scientists (i.e., faculty of local, regional, and national universities in the US) and trainees on topics arising from community needs and matching scientific and community research interests. Specifically, the Holt residents, trained as Project Advisory Council members of Project SOAR, identified environmental issues and the need for community gardens as research questions of community interest. Primary qualitative research questions arising from residents, and faculty working with residents, included: (1) what environmental factors directly influenced residents’ everyday functioning, quality of life, health and well-being, (2) how did residents cope with these issues, and (3) what recommendations did residents have to improve their environment and empower their community?

## 2. Materials and Methods

### 2.1. Site

Holt is an unincorporated community occupying 3.178 square miles in Tuscaloosa County, Alabama, United States. The population was 4413 in the 2020 census. Holt appeared on the 1930 and 1950 census; however, the community first appeared as a census-designated place in 1990. Holt experienced great devastation in the EF4 tornado in Tuscaloosa on 27 April 2011. In addition to its own storm damage, residents complained of debris from throughout Tuscaloosa city and county as a result of the tornado being dumped in their community. During the period of data collection for this qualitative inquiry (2015–2017), the demographic characteristics of Holt residents included race/ethnicity reported as 36.2% Non-Hispanic White and 55.1% African American. The unemployment rate was 6.9%.

### 2.2. Participants

The PCORI-funded Project SOAR (Contract #1097 to Allen) recruited, trained, and sustained project advisory councils (PACs) in two under-resourced communities in Alabama: one rural and one urban. Participants in each community PAC were recruited and selected by community liaisons (e.g., Spencer and Payne-Foster). Selection criteria for these individuals included: (1) residence within the designated community or status of having lived in Holt during childhood/K-12, (2) aged 30 or older, (3) ability to read and speak English, (4) absence of a primary diagnosis of severe mental illness, substance-use disorder, suicidal or homicidal ideation, (5) intent to continue to live in the community for at least one year, and (6) stable physical health with no intention to enter long-term care within the next year. Members of these PAC groups were trained to work collaboratively with local, regional, and national research partners on research questions of mutual interest [15]. Two four-hour training sessions included (1) human subjects’ ethical research training; (2) the basic elements of community-based participatory research; (3) procedures to evaluate research materials, including recruitment flyers, surveys, and intervention manuals; and (4) communication skills for interacting effectively with scientific research partners in delivering feedback regarding their research materials. The main role of each of the PAC members was to act as community research gatekeepers and to provide useful, directive, and actionable feedback to scientific partners on research of mutual interest within their communities.

Each community worked with members of the scientific team to identify research questions of interest to their specific community. Examples of topics included materials for PAC training in the Deep South in comparison with CARDS© training conducted in the Midwest [15], surveys regarding the preparation for future healthcare needs, Head Start program implementation, and cognitive functioning and community-based dementia care. As such, the current analysis focuses on Holt, the urban site, because residents identified environmental concerns as their primary area of interest (the rural site focused on diabetes prevention, assessment, and treatment). In this way, the Holt PAC members initiated this research inquiry, in accordance with typical community-based participatory research procedures [16]. Moreover, this approach draws on concepts from citizen science, the practice of involving lay people in the conduct of science. One form of citizen science is “popular epidemiology…important for medicine and society because people often have access to data about themselves and their environment that are inaccessible to scientists” [17] (p. 127).

### 2.3. Participant Demographics

Holt PAC members (*n* = 8) consisted of two African American men and six African American women ranging in age from 41 to 65 (M = 56.63; SD = 8.7) years. The Holt PAC lost one member (an older African American woman known as “the mayor of Holt”) due to significant health decline. In comparison with the total population of Holt at the time of data collection, 100% of PAC members self-identified as African American. Across the course of Project SOAR, the Holt PAC met in person at Second Baptist Church 14 times, once per month for 90 min.

### 2.4. Procedure

Three focus-group interviews and one individual interview were conducted with seven of eight Holt PAC members (87.5%) by graduate research assistants using an interview guide asking general information about demographics, including age, race, and length of residence in Holt. Additionally, interviewers asked questions regarding: (1) their knowledge of industrial and environmental history in Holt, (2) their knowledge about potential air, soil, and water pollution, and (3) other issues around environmental justice. The Institutional Review Board at The University of Alabama approved the project and informed consent, including permission to audiotape the interviews, occurred prior to data collection. Each of the focus group or individual interviews lasted approximately 18 min. Participants were thanked for their participation and expressed gratitude at having their voices heard.

### 2.5. Data Analysis

Each audiotaped interview was transcribed verbatim for analysis. For this article, the analysis focused on data regarding the effects of industrial pollution in the Holt community. The steps involved in this thematic content analysis [18] included: (1) preparing the interview data for analysis; (2) initial reading of the transcripts; (3) re-reading of the transcripts with annotations of open codes and any thoughts in the margin; (4) sorting items of interest into proto-themes; (5) examining the proto-themes and attempting initial definitions; (6) axial coding; and (7) constructing the final form of each theme. The graduate research assistant kept detailed notes as part of an audit trail [19], documenting every step of the coding process to help document analytic decisions.

Best standards of qualitative and mixed methodology that support validity are rigor, trustworthiness and an awareness of reflexivity, credibility, and believability [20,21]. In this study, we increased the trustworthiness of our findings by directly examining reflexivity, or what the coder brings to the coding of qualitative data, through the use of investigator triangulation [22]. A four-member analysis team with experience and expertise in qualitative methodology (e.g., Moore, Payne-Foster, Spears, and Allen) independently reviewed transcripts to identify emergent themes. This investigator triangulation or peer review helps to keep investigators’ interpretations in check and support basic awareness of potential bias while facilitating solid evidence for the interpretation of the data [23].

## 3. Results

The focus group and individual interviews were audiotaped with each interview transcribed word-for-word for analysis. Four general themes emerged: general pollution, illness related to pollution, community engagement and knowledge, and governmental responsibilities.

### 3.1. General Pollution

From the analysis, general pollution included comments about the lack of clean air, soil, and water in the target community. The participants’ residency in Holt ranged from lifetime to three to five years. Multiple participants detailed how the air, soil, and water quality has depreciated over time due to their proximity to the industrial plants. One participant detailed how, as a young boy, he was playing in the streets during an industrial plant’s peak emission time and “*you could actually see particles in the air and see the particles land on people’s porches and cars.*” In addition to seeing the particles and ash in the air, many of the participants noted a particular smell in the area that they believe was linked to the emission of hazardous gas.

Additionally, participants explained that, in earlier years, many residents in Holt had gardens and fruit trees; however, over the years, they could observe the physical change in the soil, stating that “*in Holt you would look for red dirt or mud and you’d find black [mud/dirt] and all different colors of dirt that were signs of possible contamination.*” In order to have healthy and flourishing gardens, residents felt that they would need uncontaminated water. The status of the water quality is one of the most important factors to the participants because they reported that they “*got to drink the water, bathe in the water, cook with the water, and if you’re going to grow a garden or make one you’re going to use the water and if the water is bad, that needs to be taken care of.*” Another resident stated: “*I can remember a time when you would turn the faucet on and the water would be kind of cloudy white and you know, you’d have to let it run a while before you know, it clears up.*” Without adequate water, residents reported the belief that they are susceptible to contamination and illnesses that could be linked to industrial pollution. This was mentioned as a particular issue for families: “*as a parent I hate that if this is what’s causing my daughter to be sick, I hate that I’m exposing her to this but what can I do?*”

Finally, residents mentioned specific events, including the 2011 EF4 tornado: “*after the tornado the Holt community has just went down, period*” and
*“I know during the tornado you had these companies dumping in Holt because Holt has a lot of land and vacant properties. They were paying people to dump on their lot…My aunt complained and even wrote letters to the Alabama Department of Environmental Management and she had a lawyer. An officer…would track the dump trucks going down there and would be in an unmarked SUV and I would see him stop those trucks from dumping.”*

Residents also mentioned issues related to specific companies, including the closure of Empire Coke: “*the environmental management closed it down due to some issues,*” and “*we knew that the foundry didn’t have filters that they were supposed to have on their equipment*.” One resident recalled one incident:
*“There was an incident at the chemical plant where they had some type of mishap that caused the whole neighborhood to become ill and people went to the emergency room and all that because they became so sick.”*

### 3.2. Illness Related to Environmental Pollution

When discussing the connection between industrial pollution and the effects on the water, soil, and air quality, participants expressed their awareness of how they believe the pollution has affected their health and those around them. Studies have supported the idea that proximity to industrial plants is often linked with respiratory illnesses. With Holt being an unincorporated industrial township, many of the participants’ family members worked at a plant for long periods of time and or were exposed to the emissions. One participant detailed how his father worked for a plant and stated that “*most of the X’s employees, [tested positive] for asbestos,*” and, in response to that, the employer acquired the proper filtration equipment and added more health benefits to prevent further issues for the employees. Many of the participants detailed their own personal battles with respiratory illnesses. For example, one participant stated that she “*would have really bad breathing problems at night and it’ll be so bad that I would be wheezing and hardly catch my breath.*” Other participants noted that asthma was a common illness in their household, especially with their children. Although they could not definitively say it was the industrial pollution that was causing the breathing problems, one participant had to move out of Holt due to the effects it was having on her son’s health. She noted that “*until I moved out of Holt, I didn’t really experience what it was like to breathe fresh air.*” Subsequently, that was the participant’s decision to leave Holt to benefit her and her families’ health. However, leaving the Holt community is often not a feasible option for many residents. They have established a life within the Holt community, with some families living there for several generations. Longstanding residence fed the desire to create a nurturing environment through a will and determination to promote improvement and community engagement.

### 3.3. Community Engagement and Empowerment

When asked what they would say to their friends and neighbors to help motivate the Holt community to come together and make a difference for health and environmental changes, the common response from the participants was more community engagement across the lifespan (i.e., from both older and younger residents) and more personal accountability within the community. One resident stated: “*my message to the Holt community is that everyone needs to get involved from small to old.*” The older residents stated that, earlier in their lifetimes, they were unaware of the harmful effects of the industrial pollution due to not being informed. Moreover, they stated that initially they were too young to understand the impact of environmental pollution. Many of the participants stated other environmental concerns: they would like Holt residents to be more active in the maintenance of their yards, taking the initiative of starting small community projects to expand and motivate more involvement, having more people in the community take on a bigger leadership role around accountability, and see more neighbors limit littering, including furniture. Many residents reported having taken action towards cleaning up Holt. For example, one participant exemplified that her aunt “*was coordinating clean ups where she would find vacant lots in Tuscaloosa and she had a whole bunch of names and phone numbers, and she would call and enlist young men who wanted to help clean up.*” Another participant would gather men from his church to help cut the grass along with the surrounding areas. The majority of the participants believed that these small community projects would help make the community environmentally better, as well as motivate the community to advocate for the government to change things and be more involved in their community.

### 3.4. Suggestions for Governmental Responsibilities

During the discussion on the topic of community engagement, participants explained that while the Holt community is actively working towards improving their environment, they would like to have more involvement of government officials taking responsibility to assist the community in the needed improvements. Some participants believed that the absence of the county commission’s attention to Holt (an unincorporated township) is noticed by both the older and younger generations. Many residents believe that the seeming lack of interest or evidence of initiating improvements in these several factors destroys trust between community members and government officials. For example, one participant highlighted a lack of trash pickup in the community by stating: “*Now trash can sit on the side of the road for not weeks but months at a time*” and “*ever since the tornado, we have not had trash pickup but once a week.*” Participants reported perceiving that government officials were frequently unresponsive to maintenance calls, including the low maintenance of the water pipes and an overall lack of getting involved in the community. Holt residents would like to see government officials intervene after the dumping of hazardous fluids in multiple vacant lots with or without consent from the landowner. Many participants stated that they only see the county commissioners’ presence during the election period when candidates put up posters and clean up the community before local elections. As a whole, participants stated the desire for municipal and community involvement in joint environmental initiatives. Environmental justice thrives on the advocacy and involvement of everyone in the community across the lifespan to create nurturing environments, implement regulations, and monitor environmental compliance. Residents in Holt would like to feel that governing bodies are receptive to their needs and their concerns are heard.

## 4. Discussion

Prior work demonstrates the relation between a higher risk of environmental toxicity and residence within communities of color or low socio-economic status. This study, however, adds meaningfully to this body of knowledge by incorporating community based participatory research feedback from trained members of a Project Advisory Council consisting of individuals from a community labeled hard to reach for research into a qualitative examination of environmental injustice. The goal of Project SOAR was to create a dialogue with the Holt community PAC, and to identify and address the most pressing issue affecting their community. Environmentalist and public health advocates have used proximity to industrial facilities as an indirect link between water/soil quality and the occurrence of respiratory and chronic illnesses [1]. While it cannot be determined that there is a direct link with whether prolonged exposure to harmful chemicals from industrial facilities can cause subpar water and soil quality, chemicals such as Cresol (mixed isomers) are found in the water in Holt, and prolonged consumption of said water can result in respiratory tract irritation, and illnesses of the gastrointestinal system, liver, and kidneys [14,24].

A hallmark of Project SOAR was the selection, training, and retention of Project Advisory Council members who were not “connected to or comfortable with research and academia” [15] (p. 70). Rather, neighborhood residents purposively shared expertise with researchers about *how* best to partner with and meet the needs of the community. In this study, residents shared their knowledge of the history of the Holt community and their first-hand experience with the effects of industrial pollution with academic researchers in order to give them both a context as well as potential solutions for collaboration. Two theoretical approaches are worth noting in reference to this study and the implementation of the Holt PAC in Project SOAR: the Consolidated Framework for Implementation Research (CFIR) [16] and public health interventions that create “nurturing environments” [25].

First, CFIR identifies and defines constructs within implementation science with the goal of providing guidance for formative evaluations to build the implementation knowledge base across settings. Five major domains comprise CFIR: intervention characteristics, outer setting, inner setting, characteristics of the individuals involved, and the process of implementation. Biglan and colleagues’ [25] conceptualization of nurturing environments was published several years after CFIR, but this concept holds promise for research and community advocacy efforts that support environmental justice. Nurturing environments are created through collaboration (e.g., intervention characteristics, characteristics of the individuals involved, and process of implementation). Biglan and colleagues state that nurturing environments may enhance physical and mental health in under-resourced communities (e.g., outer and inner settings), via: (1) minimizing biologically and psychologically toxic events; (2) teaching, promoting, and reinforcing self-regulatory behaviors; (3) monitoring and limiting problematic community behaviors; and (4) fostering psychological flexibility among residents [25]. The attention to and remediation of environmental justice concerns provide the opportunity to minimize biologically and psychologically toxic events, and the findings of this study suggest community engagement and governmental advocacy may provide one method for addressing these concerns. Training of the Holt PAC members in the evaluation and implementation of collaborative research provided them the opportunity to train and promote healthy environmental behaviors among neighbors within their community. For example, the Holt PAC subsequently designed and implemented a “Potted Plant Project” to address community needs for healthy foods regardless of soil and water pollution concerns. The Holt PAC members functioned as true community gatekeepers and monitored community-science partnerships to facilitate positive and minimize negative “town-and-gown” research implementation behaviors. Finally, the Holt PAC members developed and disseminated information to promote psychological flexibility—the ability to be mindful of thoughts and feelings and to act in the service of one’s values, even when one’s thoughts and feelings discourage taking valued action. Similar to prior applications of the nurturing environments model to promote smoking cessation and smoke-free environments [25], community-led public health interventions may directly monitor and potentially reduce environmental pollution through community engagement and advocacy.

One of the primary findings of this qualitative inquiry involved the reported absence of governmental assistance and lack of presence within the community due to its unincorporated status, contributing to a lack of perceived support for combating environmental injustices. Many residents felt that, similar to many other low-income communities of color throughout the country, the lack of maintenance of their water supply is causing the contamination of their water and air, leading to respiratory illnesses, such as asthma. Many of the study participants felt that having the status of an unincorporated township made it difficult to advocate for themselves due to the lack of governmental resources available to them. This research inquiry and CBPR efforts with trained community advocates without historic familiarity with research and academia provide promising evidence that, with training and support, community members may be empowered to identify and address environmental concerns.

As in all studies, there are limitations. In this study, only a small number of focus groups or individual interviews were conducted with trained community advocates participating in a larger community partnership with a local university. Notably, Project SOAR was not originally designed to address issues of environmental injustice; rather, this topic arose from the partnership. While this may not be a limitation of the current study since it arose in response to community interests, it is possible that important environmental issues were missed due to the parent study’s (e.g., Project SOAR’s) lack of specific and targeted scientific designs to partner with the community in addressing this issue (see CFIR principles in the design of implementation of science research [16]). Notably, in accordance with community-based participatory research principles, the Project SOAR Team took specific actions to address this emerging concern of Holt residents through partnership with scientists and other academic resource people selected as having interests in environmental justice issues.

An innovative aspect of this study was the express goal of encouraging the scientific resource people who were invited to Project SOAR PAC meetings to share their knowledge with Holt residents, and *also learn from the residents* how to be better researchers, to be more attuned to community interests. As a result, researchers reported having a better understanding of Holt’s history and a stronger orientation to key concerns of residents. Residents shared numerous examples of toxic waste and pollution from the surrounding industry. Once this theme arose from the interviews and discussions, Project SOAR leaders invited resource people to talk about other communities that had faced environmental pollution and how the residents of those communities had worked to address pollution problems. It is noteworthy that the concerns perceived by residents of Holt are supported by historical documents. Academic resource people (Spears) brought information from the Environmental Protection Agency’s Toxic Release Inventory that mapped toxic emissions in Holt and uncovered news stories about prior litigation [2]. Additionally, members of the Project SOAR Team shared environmental toxics and health information with Holt residents in collaboration with PAC members at Health Fairs held at the local high school. As a result, collaborative community-based interventions and opportunities for meaningful research arose from these referrals. The resulting community-scientist partnership established an “Eco-Health” group consisting of biological and behavioral scientists and community residents with plans to conduct testing of water quality. This initiative was interrupted by the COVID-19 pandemic.

## 5. Conclusions

It is clear that the concept of environmental justice is vital to the establishment of nurturing environments to reduce health disparities in the Holt, Alabama, community. The results of this qualitative and community-based participatory research study suggest that, although local residents strive to increase their community advocacy to improve their environment, strategies to improve governmental representation need to be explored. Moreover, the establishment and implementation of effective intergenerational partnerships, perhaps in collaboration with university-based service learning courses, in order to address the environmental concerns of residents, are needed. It is hoped that more research and community-based insight can increase awareness of environmental health disparities caused by industrial pollution. Additional scientific and community partnerships, such as Project SOAR, are needed to empower community leaders with the knowledge, skills, and tools to fight for environmental justice. Although the number of industrial businesses has decreased over time, Holt is still experiencing environmental limitations, including poor air and water quality. In order to alleviate this issue, continued community engagement and serial annual water and soil testing is needed. Additional environmental needs include the installation of new water pipes and serial monitoring of wastewater treatment. In addition to helping Holt community residents, advocacy efforts with local government officials would build trust and strengthen the community’s representation.

## Data Availability

Qualitative data transcripts from this study may be obtained from the authors.

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
