# Peer review of "Community Perspectives and Environmental Justice Issues in an Unincorporated Black Township"

_ijerph, 2022, doi:10.3390/ijerph19127490_

Round 1

Reviewer 1 Report

I commend the authors for a high-quality, thorough, and thoughtful rewrite. It reads very well now, and the order of presentation combined with the additional depth and detail make this a very satisfying paper. I particularly appreciated the clarity and detail provided in the methods--I feel like I have a much more precise understanding of the project. I also appreciated the authors' push back on my previous suggestion of adding the urban case, and agree with their thinking. This is a tidy and efficient paper worthy of publication in its current form. I have but one very minor suggestion and that is to change the first use of the word "dirt" to "soil" on line 239, and leave it as "dirt" in the quote. 

Author Response

Dear Reviewer,

Thank you so much for your positive comments on our revised manuscript. Your comments, and those of the other reviewers, have enhanced the description of our project greatly. As per your recommendation, we have made the suggested word change in our revised and uploaded document.

We look forward to publication of this work in IJERPH.

Best regards,

Rebecca S. Allen, Ph.D., ABPP

This manuscript is a resubmission of an earlier submission. The following is a list of the peer review reports and author responses from that submission.

Round 1

Reviewer 1 Report

The manuscript offers an interesting case study of community involvement to improve the health of a community by encouraging involvement in environmental enhancements. Overall the manuscript is clear and concise. The methodology described on lines 133-141 was quite good. However, some clarification of the timing and role of community members in order to accurately describe the project as “community based participatory research”. The main tenant of such projects are that community members are involved BEFORE the research begins by collaborating on a study’s goals and methodology. Participating in a study as an interviewee, or focus group member does not qualify as CBPR. The authors need to clarify whether community members participated in the creation of the interview questionnaire, or some other aspect of the study before it commenced (e.g., before interviews were conducted). The second point that requires clarification is that ill effects of drink water are most often associated with ingestion as a route of exposure and adverse health outcomes associated with the gastrointestinal track not respiratory system. Although are cases where toxic volatile organic chemicals, or the presence of the bacterium Legionella have been associated with respiratory illness, those cases are rarely discerned by water customers. Therefore, the link between poor water quality and respiratory illness needs more explanation. What did the consumer say that led to this linkage? Were there news articles that pointed to a linkage between poor water quality. Is the water purveyor listed in this or similar reports: http://adem.alabama.gov/programs/water/waterforms/2019PWSVComplianceReport.pdf ?

Minor edits:

Line 54 remain not remains

Line 95 Put “The” in front of PCORI (or spell out – don’t start a sentence with an acronym)

Line 116 Institutional Review Board should be capitalized

Author Response

  1. Regarding the need for clarification in the methods about the timing and role of the community members and whether this research could be considered CBPR, the focus on environmental issues arose from members of the Holt PAC. Pursuing the qualitative research question described in this article was a joint decision by the Holt PAC members and scientific partners in a true CBPR fashion. This procedure has been more fully and clearly described in the Methods section.
  2. The authors argue that this project is an example of CBPR. The parent Project SOAR was designed to train and empower community members to be full partners in research inquiries within their communities (see also Kaiser, Thomas, and Bowers, 2016). After training and a few initial Project SOAR PAC meetings with the scientific partners on various topics, the Holt PAC members raised questions regarding Environmental Justice as a primary topic of interest. Thus, environmental issues became a research focus within this community due to the influence of the Holt PAC members. This issue was not part of the original scope of Project SOAR.
  3. This reviewer asks that we better describe the link between poor water quality and respiratory issues. As described in the Introduction, Professor Ellen Spears met with the Holt PAC members and discussed a toxic release inventory and the history of early environmental litigation in the Holt community. The Holt PAC group arranged for volunteer water testing in the river nearby, collaborating with faculty and students in the Biology Department, but sampling errors invalidated the results. Environmentalists and public health advocates have used proximity to industrial facilities as an indirect link between water/soil quality and the occurrence of respiratory and chronic illnesses. While it cannot be determined that there is a direct link with whether prolonged exposure to harmful chemicals from industrial facilities can cause subpar water and soil quality, chemicals such as Cresol (Mixed isomers) are found in the water in Holt and prolonged consumption of said water can result in respiratory tract irritation, illnesses with the gastrointestinal system, liver, and kidney (ATSDR, EPA). These points are made in the Introduction and Discussion.
  4. We thank this reviewer for bringing our attention to minor editorial corrections; we have corrected these issues.

Reviewer 2 Report

I find the topic that you have developed in this research very interesting, but there are certain aspects that can be improved:
- The data collection took place between 2015 and 2017. Surely the information obtained does not adjust to reality.
- In the “Discussion” section, you comment: one of the main results of the research is…….; What are the other findings obtained in the study?
- Conclusions are very brief.
- I think it would be interesting if the Project SOAR Team proposes an Environmental Management Plan to the local government to prevent, control, reduce, correct and solve the negative environmental impacts of Holt.

Reviewer 3 Report

This paper deals with a project of community engagement with a university that opened questions about environmental justice. It is part of a larger ongoing project that includes another, more urban, site. At present, there isn't enough in the way of data or findings to support a manuscript in its current form.

The description of methods is vague, so it is possible I am not understanding what has been done--"interviews" and "focus group interviews" are mentioned, and it seems that these both imply the latter, and the three conducted represent a maximum of eleven people from the community (could also be seven, but why withhold that information?). One possibility that would bring this currently thin study to a level where it contributes is that the authors could use this more urban case as a comparison to the more rural one, allowing for a deeper examination of both. In that case, the limitations of a tiny sample size of interviewees (I'm not seeking statistical reliability, but the authors have not achieved any sense of thick description of the case) might be overcome by thematic comparisons between the cases.

A few specific questions/observations:

  • What is Holt like? How big is it? What are the population demographics?
  • The Bullard citation, as it is, wafts at a large body of literature. What do does the author want to do with this, or what is it doing for them? It's a thin literature review overall, and is not clear on why/how the pieces are important to the manuscript.
  • Why "nurturing environments" as a conceptual frame? How has it been used--beyond the single source--and why is it appropriate given the need for more detailed study on the influence of the physical environment? Given the importance of this concept to the manuscript there should be a fuller explanation of why it is useful, even given weaknesses. How is it better/different or how does it contribute to the EJ literature they hint at above?
  • What kind of "scientists?" This feels imprecise and rhetorically casual.
  • If part of the innovation here is some kind of SOAR/scientists/university/community cluster that is important/helpful/useful, explain more about that. Why is it different than the many other university-community partnerships? 
  • Explain more clearly your methods, particularly around the participants. Who was interviewed and how were they selected? There is almost an implication that the PAC was interviewed but I don't think they were. Was it only the focus groups? How were they identified? In what way(s) might they (not) be representative of the community as a whole?
  • Table 1 is not the interview guide suggested by the text.
  • Analytical methods are sound, but the sample is unclear even though the text is repetitive from earlier sections. 

With regard to findings, I'm left somewhat nonplussed. Each section is so brief, and underspecified--especially with the (possibly inaccurate) use of the term "many" to count interviewee responses. It appears the whole sample barely makes it to "some," much less "many." My advice here would be to Strunkify the description--be specific when you can. If it is correct that you have only three focus groups of about ten people total, and the findings of those groups are not particularly new, then it would seem to me that the contribution to be made would be to compare with your urban case and expand the focus of the paper. 

In the conclusion, the authors suggest this is community-based participatory research. Per my suggestion to detail SOAR, this claim should be more fully detailed. As it stands, the interviewing doesn't seem like it meets the standards of CBPR. 

I think community-based opinions about EJ issues are very worthy of reporting, and so encourage the authors to consider necessary revisions: use the literature more incisively, clarify their methods and sample data, and expand the manuscript to detail why SOAR is important and anchor these findings in that larger discussion, perhaps necessitating the inclusion of the other rural case. 

Reviewer 4 Report

This is a very interesting investigation. In “southeastern” of US, low-income and predominantly Black communities bear the brunt of the environmental consequences while frequently remaining stagnant economically. The authors performed this qualitative, community-based participatory research study, focused on Holt, Alabama, an unincorporated community in the southeastern United States.

One of the primary findings of this qualitative inquiry involved the reported absence of governmental assistance and presence within the community due to its unincorporated status, contributing to a lack of perceived support for combating environmental injustices.  Many residents felt that, like many other low-income communities of color throughout the country, the lack of maintenance of their water supply is causing the contamination of their water and air, leading to respiratory illnesses such as asthma. Many of the study participants felt that having the status of an unincorporated township made it difficult to advocate for themselves due to the lack of governmental resources available to them.

This study was designed well and the manuscript was written well. Fighting for environmental justice is also an important topic for everyone in every country. I suggest the publication of this study.

Author Response

The authors thank this reviewer for their supportive comments and positive perceptions of our work.